# Face Transplant: Indications, Outcomes, and Ethical Issues—Where Do We Stand?

**DOI:** 10.3390/jcm11195750

**Published:** 2022-09-28

**Authors:** Simone La Padula, Rosita Pensato, Chiara Pizza, Edoardo Coiante, Giovanni Roccaro, Benedetto Longo, Francesco D’Andrea, Francesco Saverio Wirz, Barbara Hersant, Jean Paul Meningaud

**Affiliations:** 1Department of Plastic and Reconstructive Surgery, Università degli Studi di Napoli Federico II, Via Pansini 5, 80131 Napoli, Italy; 2Department of Plastic, Reconstructive and Maxillo Facial Surgery, Henri Mondor Hospital, University Paris XII, 51 Avenue du Maréchal de Lattre de Tassigny, 94000 Créteil, France; 3Department of Plastic and Reconstructive Surgery, Università di Roma Tor Vergata, Viale Oxford 81, 00133 Roma, Italy; 4Plastic Surgery Clinic, 12 de Rue de Ponthieu, 75008 Paris, France

**Keywords:** face transplantation, vascularized composite tissue allotransplantation, facial allotransplantation, facial disfigurement, decellularization, chronic rejection, face transplantation outcomes, ethics, regenerative medicine

## Abstract

Background: The addition of face allotransplantation (FT) to the head and neck reconstructive surgery arsenal has started a true revolution. This study is aimed at providing an extensive analysis of the current practice of composite tissue allotransplantation. Moreover, a thorough description of pre-procedural, intra-operative, and post-procedural settings, indications, contraindications, outcomes, ethical considerations, and future perspectives is provided. Methods: The authors’ experience was supplemented with a literature review performed by using the PubMed, MEDLINE, and Embase databases on 21 February 2022. The search terms used were “face transplantation indications”, “face transplantation complications”, and “face transplantation ethical issues”. Results: The most recent achievements and long-term clinical sequelae of FT are classified and summarized. A large number of records (4435) were identified. Seventy-five articles were assessed for eligibility. Publications without new data and reports with a patient follow-up < 5 years were excluded. Nineteen articles met the criteria for inclusion. Conclusions: The most recent achievements in the field of FT may be combined with cutting-edge regenerative medicine procedures and innovative immunological processing. It is paramount to build strong international networks between the world FT experts in order to achieve higher-level outcomes and reduce the complication rate. Nevertheless, the utmost caution is required in patient selection, clinical assessment, strict follow-up, and rejection management.

## 1. Introduction

The first face allotransplantation (FT) was performed in 2005. It initiated a true revolution in the head and neck reconstructive surgery field [1]. Since then, a new ideal standard of care has been set for patients suffering from extensive and complex facial trauma and tissue loss. FT provides very satisfying outcomes in terms of functional facial reconstruction after the failure of autologous tissue-based reconstructive techniques. Progress has been made and multiple successful procedures have been conducted all over the world, and new refined techniques have been developed. Over the past two decades, 48 FT procedures have been carried out in 46 patients (2 documented retransplantations) [2,3]. However, the annual rate of accidents leading to complex facial trauma is still extremely high [4]. Many of these trauma victims may not be eligible for autologous procedures and could benefit from FT. Vascularized composite allotransplantation (VCA) is an extremely complex procedure that relies on highly skilled and accurate dissection and insetting techniques and requires a multidisciplinary team to maximize the success rate. This article is intended to address multiple issues regarding face transplants. To date, a solid consensus among world experts is still to be developed. In addition, data regarding long-term postoperative results are few. This study is aimed at providing an extensive analysis of the current practice of FT.

Since the FT procedure has not yet been standardized, a scoping review was performed to systematically map the research conducted in this area and identify any knowledge gaps. The following question was formulated: What is known from the literature regarding pre-procedural, intra-operative, and post-procedural settings, indications, contraindications, outcomes, ethical considerations, and future perspectives?

## 2. Materials and Methods

The authors’ experience was supplemented with a literature review [5] using the PubMed, MEDLINE, and Embase databases on 21 February 2022. The search terms used were “face transplantation indications”, “face transplantation complications”, and “face transplantation ethical issues”. Only articles reporting patient follow-up > 5 years were included. Exclusion criteria were a patient follow-up < 5 years and articles with no full text available.

To increase consistency among reviewers, all authors examined the same publications and discussed the results. They modified the screening and data extraction manual before starting screening for this review. Ten reviewers evaluated the titles, abstracts, and full texts of all papers to identify potentially relevant ones. Four reviewers together developed a data-graphing form to determine which articles to extract and use for this review.

## 3. Results

The most recent achievements and long-term clinical sequelae of FT were classified and summarized. A huge number of records (4435) were identified through PubMed, MEDLINE, and EMBASE. A total of 125 records were screened. Seventy-five articles were assessed for eligibility. Publications without new data and reports with a patient follow-up < 5 years were excluded. Nineteen articles met the criteria for inclusion (Figure 1 and Table 1).

Data obtained from the 19 included articles and additional interesting papers [1,2,3,4,5,6,7,8,9,10,11,12,13,14,15,16,17,18,19,20,21,22,23,24,25,26,27,28,29,30,31,32,33,34,35,36,37,38,39,40,41,42,43,44,45,46,47,48,49,50,51,52,53] were organized and are reported in the following paragraphs.

### 3.1. FT: Surgical Indications

The main indications and patient selection are extremely strict. The inclusion criteria were established by a multidisciplinary team including plastic surgeons, psychological therapists, speech therapists, dentists, and transplant specialists. This team is expected to extensively appreciate and evaluate the patient background and social network and support, which is crucial for a successful recovery, psychological resilience, and compliance with life-long immunosuppressive treatment. Immunological risk factors also need to be taken into account when undertaking FT. Patients with a past medical history of burns and transfusions may have developed immunosensitization. This condition may complicate donor selection and lead to poor post-procedural outcomes [6,7,21]. In addition, life-long immunosuppressive treatments may also lead to an increasing incidence rate of de novo malignancies. Major facial trauma causing significant tissue loss to the middle third of the face and extensive damage and disruption to crucial and functional anatomical structures such as the nose, lips, and eyelids is the most widely recognized indication for facial transplantation. In particular, total destruction of the orbicularis oris and of the orbicularis oculi muscle and total face destruction represent the limit of classic reconstructive surgery. Patients sustaining ballistic trauma, burns, and animal bites most commonly undergo FT [8]. FT has also been carried out in patients suffering from extensive facial plexiform neurofibromas or significant facial disfigurement after cancer excision [9,22,23].

#### 3.1.1. What Kind of Patient Can Really Benefit from Facial VCA?

Conventional techniques can treat most facial injuries: the destruction of the nose, jaws, cheeks, etc. However, beyond a certain stage of severity, some loss of substance cannot be functionally or aesthetically repaired [2,3,4,5,6,21]. At the current stage of our knowledge, it is not possible to repair orbicularis muscles (lips and eyelids) when they are totally destroyed. We are only able to repair these circular muscles when they are partially lost by muscle transfer and duplication (to lengthen them), but in order to be able to repair these structures, it is necessary to have enough residual muscle [9,22,23]. To achieve minimal results, these patients may undergo numerous operations. However, beyond the cumulative risk of all these conventional procedures, the outcome is always disappointing, both functionally and aesthetically. A common mistake is to think that the face is only a superficial plastic entity that allows us to recognize ourselves. This aspect should not be neglected because a minimum appearance is required to perform most social functions. Beyond a certain degree of disfigurement, social life is seriously impaired. However, the face is more than a social interface. In fact, the face must be considered an organ. It enables or facilitates functions as basic as eating by mouth, salivating, chewing, swallowing, preventing burns, and avoiding drool, and it allows kissing, protection of the eyes, breathing through the nose, warming the inspired air, and speaking. In practice, a patient who is a candidate for FT has, above all, a functional requirement. Most patients with severe facial disfigurement cannot perform any of the functions cited above and are sentenced to tracheotomies, gastrostomies, permanent drooling, and silence. In short, to establish the indication for a face transplant, a circular muscle must have been completely destroyed. It may be the orbicularis muscle of the lips, the orbicular muscles of the eyelids, or, of course, the whole face [24].

#### 3.1.2. What Kind of Pathologies Can Lead to Facial VCA?

Five pathologies or types of traumas are essentially associated with face transplants: ballistic traumas, neurofibromatosis, animal bites, burns, and certain sequelae of cancer [10,11,12,13,14,15,16,25,26,27,28,29,30,31,32,33].

#### 3.1.3. The Cancer Sequelae Problem

FT after cancer resection should not be indicated. As a matter of fact, immunosuppressive treatments increase the risk of recurrence. In 2009, in Valencia, Spain, a face transplant procedure was carried out on a patient with a past medical history of HIV infection and the radical excision of a vast buccal mucosa squamous cell carcinoma [21]. Despite promising early post-procedural outcomes, the patient suffered tumor recurrence leading to death. In 2003, in Nanjing, China, a 72-year-old female patient underwent wide resection of locally advanced and metastatic melanoma (lymph node metastasis). The resulting defect was resurfaced through scalp and bilateral ear allotransplantation [34]. The patient died 6 months later due to metastasis. Based on previous clinical reports, patient selection should rely on extremely strict criteria. Past medical history should be closely analyzed and taken into account. The potential risks of the procedure should never outweigh the benefits. Immunosuppression in a patient with a history of malignancy will increase the risk of life-threatening complications, shifting the balance towards unacceptable risks. The risk of de novo malignancy is 2- to 4-fold higher in immunosuppressed recipients [35]. However, devastating composite facial defects resulting from malignant tumor resection represent an absolute contraindication to FT, a past medical history of malignancy is a relative contraindication, and a history of benign tumors represents a strong indication for a face transplant. The ongoing research on immunosuppressive medication (modulation of immunosuppressive effects) may extend face transplant applications to patients suffering from severe deformities resulting from malignant tumor resection [36]. However, FT may be offered to patients who have had maxillofacial cancer at a relatively young age and are undergoing the sequelae of radiation therapy (RT) more than twenty years later. Radiotherapy has a progressive effect over time that can lead to loss of function due to tissue retraction, which is very debilitating and sometimes impossible to treat. The number of patients who would fall under this indication is probably important, but, technically, these are probably the most difficult cases because recipient vessels may have been damaged by RT.

### 3.2. Preoperative Considerations and Surgical Procedures

#### 3.2.1. Donor Selection

Donor selection and matching in facial VCA is very challenging and more complex than in solid organ transplantation. The donor and recipient must be matched appropriately based on blood type (ABO group) and immunologic criteria (HLA), as well as demographic factors, hair and skin color, and cephalometric characteristics. These considerations have made the donor shortage more pronounced in facial VCA and have often resulted in extended waiting times for candidates before FT [30,31,32,33]. Concerning the donor gender, in theory, it would be possible to graft tissues from a woman to a man without this posing any problems because they are under hormonal influence. However, in practice, the sexes have been concordant to date [31]. A more important criterion is that of size. It will not be possible to graft a small face onto a large head. Strong collaboration between transplant centers and organ procurement organizations can reduce waiting times for facial VCA candidates by expanding service areas for donations [24].

#### 3.2.2. Recipient Preparation

It is essential that the patient is properly balanced at the psychological level. Most often, disfigurement requires psychotropic treatment and psychological support that may take several months or years [33]. A climate of trust must be established with the whole team because the transplant will be an ordeal for the patient, and it is therefore necessary that exchanges can take place with respect and dignity, regardless of the patient history, which is inevitably particular. A complete clinical and paraclinical assessment (biological and radiological) has to be carried out. This includes HLA grouping, the search for antibodies, and the realization of serologies for the human immunodeficiency virus (HIV), hepatitis, cytomegalovirus, and herpes. A CT angiography of supra-aortic axes is required in most cases to study the recipient’s vascular axes. According to some FT teams, a conventional digital subtraction angiography may be required [2]. Electromyography (EMG) is performed to assess residual motor (facial nerve) and sensory (trigeminal nerve) functions [9,10,22,23,24]. A search for infectious foci is carried out with bacteriological examinations and dental, otorhinolaryngology (ORL), and pulmonary check-ups. Due to the anti-rejection treatment, the patient’s immune defenses will be depressed, and infectious foci that are not treated beforehand could endanger the patient’s life [24].

#### 3.2.3. What Does the Procedure Consist of for the Donor?

It is suitable that a face harvest is performed first (before other organs are harvested) under good hemodynamic donor conditions to avoid high levels of catecholamines that may have a possible detrimental effect on endothelial surfaces (during organ harvesting, catecholamines are administered to maintain adequate blood perfusion; doses may vary depending on the patient’s condition) [10,11,12,13,14,15,16,24,25,26,27,28,29,30,31,32,33,34,35,36,37]. The first step is to remove all probes connected to the face (venous access, gastric tube, etc.) and perform a tracheostomy. It is necessary that the entire face can be accessed by the surgeon without any interference. Then, a cast of the face is made for the design of a resin mask that allows, at the end of the harvest, the body to be returned to the family in conditions that respect human dignity.

Various methods of donor face restoration are available today, most of which include different materials and molding techniques to produce donor masks to restore, to the greatest extent possible, the preoperative appearance of the donor’s face. Alginate is usually used to create a mold and negative impression of the donor’s face; this process requires approximately 30 min to complete. Once the alginate impression is completed, mask production continues in a separate room with colored acrylic resins that are subsequently poured into the mold. The mask is then perfected with the application of makeup by a maxillofacial prosthetic technician or anaplastologist using a photograph of the donor. At the end of the harvesting of the facial VCA, the mask is placed on the donor under the surgeon’s supervision. The average production time for the masks is about 4 h, and the cost of materials is $50 per mask [14].

Depending on the trauma of the recipient, harvesting surgeons prepare a varying number of anatomical structures. For example, if the trauma of the recipient patient concerns the lower part of the face, often the lower two-thirds, the harvesting surgeons will take the corresponding tissues “en bloc” with respect to the anatomical units. This means that to avoid the effects of “patch” or “patchwork”, it is preferred to rebuild a whole structure rather than a half-structure, for example, a whole cheek rather than a half cheek. For lower-face harvesting, both facial nerves, the orbicularis muscle of the lips, the second and third branches of the trigeminal nerve, the toothed part of mandibular and maxillary bones, the main salivary glands with their ducts, the nose, and, of course, all superficial cutaneous and mucosal structures are harvested. The whole transplant must be pedicled on the external carotids and jugular veins at a level where their caliber is as large as possible. The harvesting of the superior third of the face includes the eyelids with the orbicularis muscle and the eyelid levator muscle, the tear ducts, the conjunctiva, and possibly the lacrimal gland. Temporal vessels must be harvested to ensure the vascularization of the upper part of the face [2]. An upper-face harvest is more complex than a full-face harvest. Indeed, the upper-face harvest is a full-face harvest from which the useless parts are secondarily isolated. Anatomically, the scalp and the external ear are not a part of the face but can be included in the harvest. The harvesting surgeons work at the same time as the “recipient” surgeons. They must coordinate to limit the time of graft ischemia as much as possible. At the end of the face harvest, surgeons will rinse the graft and inject it with a preservation solution.

Static cold storage of the VCA and preservation solutions are used to reduce the risk of ischemia–reperfusion injury. Preservation solutions are differentiated based on their Na+/K+ ratio. High sodium is a characteristic of an extracellular-type solution (ETS), while high potassium corresponds to an intracellular-type solution (ITS). ITS maintains the intracellular ion concentration by creating a temporary cation equilibrium that compensates for the lack of active transport during ischemia. While many different preservation solutions exist, the University of Wisconsin solution (UWS) has been used in most cases of facial VCA. This is an ITS solution with a pH of 7.4 and an osmolarity of 320. It contains 25 mmol/L sodium, 125 mmol/L potassium, 25 mmol/L phosphate, 5 mmol/L sulfate, 5 mmol/L magnesium, 30 mmol/L raffinose, 100 mmol/L lactobionate, 5 mmol/L adenosine, 50 mmol/L pentafraction, 3 mmol/L glutathione, and 1 mmol/L allopurinol.

The graft will then be kept in a cooler specially designed for transporting samples and delivered as quickly as possible. Currently, it is rare that the total ischemia time exceeds four hours [24].

#### 3.2.4. Operation on the Recipient

Operations on the recipient and the donor start at the same time. The first part of the procedure on the recipient patient should be exposure of the recipient's vessels and nerves, and then removal of scar tissue, i.e. tissues that have no function. In the case of neurofibromatosis, this stage can be particularly difficult because of the bleeding. More than elsewhere, improvisation may expose the patient to massive hemorrhage and death. Recipient vessels (external carotid artery and venous trunks) and recipient nerves (trunks of the facial nerves and branches of the trigeminal nerves) are identified and prepared [2,3,4,5,6,7,21]. This step is difficult when patients have been operated on multiple times or when it comes to burns. The graft is delivered in a refrigerated box. After being flushed, arterial anastomosis is carried out. Next, venous anastomosis is performed, followed by contralateral anastomoses. It is then necessary to adapt the graft to the loss of substance of the recipient patients. This is a step that can be difficult in the case of the loss of bone structures because there is often an incongruity. Bone structures are connected using titanium screws and miniplates (osteosynthesis). The main difficulty resides in the fact that once this osteosynthesis has been performed, certain structures that must be sutured, such as the palatine veil or certain nerves, will no longer be accessible. The difficulty is circumvented thanks to a set of wires placed in advance, which will be lowered at a distance according to the slipknot principle. Finally, mucosal and skin sutures are performed [24].

### 3.3. Facial Defect Classification System for FT

The vast potential of FT encourages communication between international teams to try to standardize this procedure as much as possible. In FT preoperative planning, Le Fort–type osteotomies and functional and aesthetic soft-tissue facial subunits should always be taken into account to allow for correct facial allograft harvesting and recipient tissue excision. Soft-tissue and skeletal tissue defects for a facial allograft have been classified by Eduardo D. Rodriguez et al. [2]: *Soft-tissue defect classification:* type 0, oral defects (upper and lower lip and commissures); type 1, oral–nasal defects (nasal soft-tissue architecture defect with or without type 0 defects); type 2, oral–nasal–orbital defects (infraorbital and malar tissue with or without type 1 defects); and type 3, full facial defects (all facial soft tissues). *Bone tissue defect classification:* type A, Le Fort I type (partial or total maxillary defects); type B, Le Fort III type (maxillary defects, inner and lower orbital borders, and zygomatic bones with or without nasal, vomerine, and ethmoid bone tissues); type C, monobloc advancement type (frontal and supraorbital bones with or without bone tissue involved in the other types); and subtype M, partial or total mandible involvement (Figure 2).

### 3.4. Immunosuppressive Treatment

Immunosuppressive treatment begins as soon as the FT is completed with an induction treatment that generally includes three or four drugs: anti-thymocyte globulin (ATG) or anti-IL-2 receptor antibody in combination with tacrolimus, mycophenolate mofetil (MMF), and steroids (prednisone). The induction treatment will be continued with maintenance treatment, the terms of which vary according to teams but whose doses are much lower [2]. Maintenance therapy traditionally consists of triple therapy with corticosteroids, MMF, and tacrolimus, although some teams have used dual therapy with MMF and tacrolimus. The technical and anatomical aspects of facial VCA can lead to infectious complications. Specifically, the flora colonizing the donor’s oral and respiratory mucosa, peripheral ganglia, and lymph nodes is transferred during the procedure and predisposes recipients to donor-derived infections. Furthermore, immunosuppressive therapy increases the recipient’s risk of opportunistic infections [3]. The flora most frequently found in nasal cavities is represented by Corynebacteriaceae, Propionibacteriaceae, and Staphylococcus, while Prevotella, Streptococcus, Veillonella, and Neisseria genera are more commonly found in oral spaces. Perioperative antimicrobial prophylaxis at our institution includes vancomycin, cefazolin, and micafungin for at least 4 days after surgery. The perioperative antimicrobial regimen is then adjusted based on donor and recipient cultures obtained at the time of transplantation. Pneumocystis prophylaxis with trimethoprim–sulfamethoxazole and cytomegalovirus (CMV) prophylaxis with valganciclovir are given for 6 months post-facial VCA, or longer if VCA recipients receive treatment for acute rejection.

One of the challenges of the current research aims is to adapt immunosuppressive treatment to what is strictly necessary. It is possible that patients are actually overtreated. The ultimate goal is to have the facial VCA accepted without the need for anti-rejection treatment. Some teams also transplant donor bone marrow cells with this hope [38,39]. There are encouraging data, but it is still too early to draw conclusions.

### 3.5. The Importance of Rehabilitation

The rehabilitation of FT recipients is essential and should start as soon as possible. Rehabilitation efforts for these patients require a comprehensive treatment approach to address their unique and multidimensional problems. The interdisciplinary rehabilitation team consists, in most cases, of physical therapists, speech-language pathologists, respiratory therapists, and nutritionists. Initial rehabilitation in the acute phase (1–30 days after surgery) focuses on helping patients achieve medical stability, recovery, and healing and includes basic activities, such as respiratory function, the ability to eat/drink, and sleeping comfortably. Patient education begins with the first contact with the patient and his or her family members. Patient and family education is continued throughout the rehabilitation process. To prevent graft failure, wound dehiscence, and infection, movement may be restricted for at least 2 months after surgery. The simplest self-care skills (protecting the face, washing the face, combing the hair, scratching the face, and blowing the nose) become a focus. The rehabilitation of the muscles and functions involved in facial expression should be one of the therapist’s main goals. It is also important to constantly check the range of movement and skin sensation in the head and neck region. From one to three months after surgery (subacute phase), static and dynamic facial exercises to improve lip suspension and mouth occlusion are an essential part of daily treatment, as well as some intervention techniques for visual processing and olfactory training. In the subsequent period (4–6 months after surgery and thereafter), when the patient progresses and is discharged from the hospital, home exercises are crucial.

Therapists must ensure that the patient becomes more independent, with functional performance including safety, mobility, the activities of daily living, visual processing, safe olfactory sensation training, speech, and swallowing. Sensory stimulation techniques and muscle-strengthening exercises are then prescribed, and surface EMG or mirror training can be used to train the movement of individual muscles. Mirror feedback, passive stretching, and massage can be used to reduce muscle hypertonicity if the patient develops muscle hypertonicity or facial asymmetry. One of the most effective tools for neuromuscular reeducation of the face is EMG biofeedback with surface electrodes. It provides real-time visual and/or auditory feedback information to the patient (while attempting to relax hypertonic muscles), increases the force of targeted muscle contraction, prevents undesired muscle activity in adjacent areas, or decreases or eliminates synkinesis patterns. The rehabilitation of the ability to smile takes place through the reeducation of muscles and motor control. The zygomatic and risorius muscles are targeted, and the patient can practice smiling, often referring to exercises in the mirror. For olfactory reeducation, the patient is usually exposed to various familiar and natural odors inside bottles of similar color so that the odor cannot be identified from the bottle type.

The principal goal of FT is the resumption of speech. Many of these patients had a tracheotomy several years before the FT, an almost immobile tongue (ankyloglossia), and the impossibility of closing the lips or, on the contrary, of opening the mouth. Speech training and swallowing training should be performed along with facial muscle therapy. Finally, psychologists play a key role in the rehabilitation and social reintegration of these patients. They help the patients recover self-confidence and a social life.

#### Functional Recovery

When the sensory nerves are connected, the resumption of sensitivity is quite fast. From the fourth month, the patient requires local anesthesia during skin and mucosal biopsies (for rejection monitoring). This happens in a short time, and it is therefore quite surprising. The immunosuppressive treatment (tacrolimus) would be at the origin of this happy side effect. In the case of impaired nervous connection, there is a sensory recovery of lower quality, allowing a certain sensitivity (proprioception). Motor recovery is fundamental. It would not be conceivable to perform an FT without proper functional recovery [16,33,34,35,36]. The onset of motor recovery occurs around the sixth month post-FT. Gradually, the patient imprints his or her own expression onto the transplanted face. Wrinkles appear with more or less acuity, such as nasolabial folds. The patient appropriates the graft. Neurological proof of motor recovery is performed through clinical observation. When the patient displays a broad smile, there is no longer any doubt. The objective estimation of motor recovery can be performed with electromyography [24].

### 3.6. Immunological Rejection

Rejection of the graft is a major complication of this procedure. On clinical examination, if erythema and edema occur, skin and mucosal biopsies of the facial VCA should be performed. The extent of the epithelial damage and inflammatory cell infiltration can vary [40]. Biopsies of allotransplanted mucosa seem to show more prominent inflammatory changes when compared to those observed in cotransplanted allogeneic skin. It is important to distinguish three forms of FT rejections. They are characterized by their time of onset, more or less early after the FT. Hyperacute rejection: it occurs in the hours that follow FT and manifests in the form of an infarction of the graft. Acute rejection (AR): it occurs starting from the fourth day post-transplant. The transplanted organ is the site of infiltration by immunocompetent cells. Chronic rejection (CR): this is the main cause of an FT failure.

The most frequent clinical signs of AR involving the skin are erythema and edema of the facial VCA. AR of the oral and nasal mucosa often appears with ulcerations and erosions, accompanied by diffuse erythema.

Recent reports suggest that, on a microscopic level, rejection occurs even more frequently in mucosa when compared with the skin.

Generally, the surface epithelium of the mucous membrane is involved, and a multifocal lymphocytic inflammatory cell infiltrate, intercellular edema, lymphocytic exocytosis, basal cell vacuolization, and keratinocyte apoptosis can be observed.

Similar to the skin, cell infiltrates largely consist of CD3+/CD4+ cells and fewer CD3+/CD8+ cells. FoxP3+ cells have been observed in both mucosa and skin biopsies (10%–15% of the infiltrate).

There was no difference in the dendritic cell and FoxP3+-lymphocyte density between the mucosa and skin. Additionally, the focal expression of HLA-DR antigens in the mucosal epithelium was detected in specimens in cases of rejection [54]. AR reactions are graded based on a scale ranging from 0 to 4 [40]. FT recipients should be accurately monitored and examined. Acute rejection should be promptly treated. The treatment of rejection episodes, typically with pulse-dose corticosteroids and the adjustment of maintenance immunosuppressants if needed, must be initiated early. In FT, episodes of acute rejection are almost constant. If there is a suspicion of AR, mucosal and skin biopsies should be performed at predefined time intervals. With a face transplant, one advantage over internal organs is that rejection can be detected earlier quite simply by observing the inflammation of the integument. The reaction time is therefore shorter. However, it must be confirmed by biopsy and microscopic analysis. The Banff score [40] allows the assessment of the intensity of the rejection and adjustment of the anti-rejection treatment that must be implemented. Chronic rejection leads to the loss of the architecture of the graft, which gradually becomes the seat of fibrosis, leading to the progressive loss of function of the transplanted organ.

In general, graft loss as a result of CR occurs years after the transplant, but the loss can occur much sooner if vasculopathy is a major component. A recent study found that graft vasculopathy has occurred in 6% of all human VCA recipients [55]. To date, six cases of CR have been reported. Two of these patients died of malignancy [2], and two patients received a second VCA. Although reports are beginning to emerge, the timeframe for CR remains unknown, and expert consensus on CR management and allograft failure has not been established. Clinical findings range from early fibrotic changes such as facial skin thinning and/or accelerated wrinkling, telangiectasia, dyschromia, and skin sclerosis with allograft dysfunction to frank necrosis. In the event of allograft loss, surgical salvage strategies are necessary, and both autologous reconstruction and retransplantation have been reported [2,3]. Kaufman et al. recently summarized CR [55] in four stages (0 to 3) based on the severity of the clinical and pathological features:CR0: Absence of any clinical or histological evidence of CR.CR1: Histological evidence (capillary thrombosis and loss of microvasculature, hair follicle apoptosis, tertiary lymphoid organ-like follicles, skin atrophy and adnexal loss, fibrosis, and vasculopathy/intimal proliferation/hypertrophy) of CR without external evidence or functional decline of the VCA.CR2: Clinical and histological (skin atrophy and fibrosis with adnexal loss) evidence of CR in the absence of the functional decline of the VCA.CR3: Overt functional decline of the graft with external and histological evidence of CR (skin necrosis, purpuric skin lesions/bruising, adnexal loss, and pain).

In the case of hand transplantation, complete rejection would bring the patient back to its initial state, but in the case of a face transplant, it would be extremely dangerous because it would lead to a state much more serious than the initial state, potentially fatal. In practice, one of the most serious dangers that await the transplant recipient soon after FT is infection. Shortly after the transplant, the risk of infection is very high. In the medium term, the infection risk stabilizes. In addition, anti-rejection treatment can be decreased so that the patient can conduct almost a normal life [24].

HLA sensitization should also be taken into account. Burn victims are usually HLA-sensitized [41]. A past medical history of trauma increases the likelihood of HLA sensitization (due to transfusions and skin allografts). Sensitized patients carry an increased risk of rejection. Higher doses of immunosuppressive therapy are therefore required. The increased rate of de novo malignancy and the opportunistic infection rate are the main complications of high-dose immunosuppressive treatment. HLA sensitization and past medical history of a psychiatric condition may compromise the success of an FT procedure. Despite immunosuppressive treatment, the long-term survival of the facial allograft might be compromised by chronic antibody-mediated rejection (CAMR). This condition can cause complete necrosis of the VCA. Unlike complete rejection in hand transplantation, which can be treated with amputation and a return to the near-pretransplant condition, the complete loss of facial allografts can be life-threatening. The first facial retransplantation in a man, 8 years after his first facial transplantation, was recently successfully performed and reported by Lantieri et al. [17]. This patient experienced the complete loss of his first allograft due to severe CAMR. Thus far, according to Lantieri [17], retransplantation appears to be technically feasible and necessary in the case of VCA loss due to chronic rejection. We visited this patient in 2017, a few months before retransplantation, since he was followed by the Immunology and Nephrology teams of the Henri Mondor University Hospital—Créteil (where he received his first FT by Professor Lantieri and Professor Meningaud). We observed diffuse telangiectasia, dyschromia, and skin sclerosis with allograft dysfunction (limited facial motion).

The patient underwent facial retransplantation in January 2018 at the Georges-Pompidou European Hospital (Paris). Although the patient had a complicated postoperative course with numerous immunological, infectious, cardiorespiratory, and psychological events, he was discharged after a hospital stay of almost 1 year. He has since been able to re-integrate into his community with the acceptable restoration of his quality of life.

In 2021, Pomahac published the second case of facial retransplantation in a facial VCA recipient following irreversible allograft loss 88 months after the first transplant [18]. CAMR and recurrent cellular rejection resulted in a deteriorated first allograft, and the patient underwent retransplantation. There was no hyperacute rejection in the immediate postoperative phase, and the outcome 6 months postoperatively seems to be promising. However, secondary FT carries an increased risk of early rejection. This is due to an immunological mechanism of sensitization to the primary allograft. In addition, fibrosis of the recipient site may increase the complexity of the procedure, and the recipient’s vessel architecture may be altered due to previous anastomoses. Follow-up reports about this case will provide further information on the long-term complications of secondary FT procedures.

### 3.7. Long-Term Outcomes and Deceased Patients

Long-term FT clinical sequelae have been described [14]. We noted, especially in patients who faced multiple acute rejection episodes, a lymphedematous aspect, tooth loss, sagging, asymmetry, loss of laxity, and pigmentation changes in the transplanted area. Personal observations of these clinical changes over time in FT patients and reported literature data allowed us to summarize some of these sequelae, as shown in Figure 3. Furthermore, in some cases, premature aging of the transplanted face has been observed, probably as a consequence of immunosuppressive therapy. All of these sequelae may often require revision surgeries.

To date, of the 46 patients who have received an FT, 8 have died. The causes of death and data of these patients are summarized in Table 2.

### 3.8. Ethical Issues

Is it ethical to perform an FT on subjects who attempted suicide? The last author answered this question in his book [24]: The honor of medicine is to treat everyone without moral consideration concerning the patient’s past, especially when the origin of the trauma is linked to psychological suffering that has reached its peak.

An accurate psycho-social analysis is crucial in patients suffering from self-inflicted injuries and in those with a past medical history of drug abuse and self-harm. Case reports of FT procedures carried out on this group of patients showed satisfactory outcomes and high success rates. However, an accurate pre-procedural psychiatric assessment is crucial to reduce the risk of self-harm recurrence [8]. The indication of FT procedures in visually impaired patients is still being discussed by global ethical committees and represents a highly controversial issue [2]. Blind patients who receive a successful FT (as with all other patients) will be able to recover facial sensations, and they will be able to speak, eat, kiss, and breathe normally again. None of this should be denied to anyone.

To date, pediatric facial allograft has never been performed. During an international ethics meeting, 62% of the attendees agreed on the indication of pediatric facial allograft in selected cases. However, multiple issues, such as parental consent, the risk of immunological complications, and the evolution of facial features during puberty, should be taken into account and weighed against the potential benefits. Moreover, multiple concerns exist regarding donor selection, informed consent, and compliance with immunosuppressive treatments [42].

## 4. Discussion

In this scoping review, we identified 30 primary studies reporting important information about FT.

Compared to recently published articles on the same topic, our review aimed to present more technical aspects and different issues concerning VCA. Thanks to the contribution of the senior author (JPM), who is a pioneer and major contributor in the field of facial allotransplantation, we have tried to describe the whole procedure in detail. Our findings indicate a paucity of research focusing specifically on the dissemination of knowledge on the FT procedure and a limited number of studies on its implementation in this area and how to address the rejection problem. A facial composite allograft is an exquisite and complex reconstructive option for patients with extensive facial disfigurement who are not amenable to autologous reconstructive approaches, providing very satisfactory functional and cosmetic results. We now have evidence that olfactory and eating abilities were restored in almost all reported cases; the ability to breathe, talk, and control facial expressions also significantly improved in FT patients [19]. Candidate selection, a multidisciplinary team approach, psychiatric assessment, and follow-up are paramount for success in FT. To date, all FT recipients and their families have been satisfied with the procedure. However, post-procedural quality of life and social interactions were significantly variable, depending on the pre-procedural background and psychiatric comorbidities. However, there is still a long way to go. Offers to patients remain limited due to lifelong immunosuppression. To date, at least six cases of CR have been reported, and nearly all FT recipients have had at least one incidence of acute rejection (AR) [3]. Like in other types of VCA, chronic rejection [20,43] induces distal vascular impairment, which results in a mean graft survival of 10 to 15 years. Fortunately, along with FT, surgical science advances further every day and everywhere, even where we least expect it. Premanufactured models of autologous orbicular muscles are being studied, and regenerative medicine now is quite different from that of 2005 [44,45,46,47,48,49]. To move forward in the field, a new pathway in tissue engineering has been proposed. In regenerative medicine, so-called cell seeding on scaffold technology (CSST), in which cells from a patient can be grown on extracellular matrix (ECM) scaffolds to produce functional organs, has been previously investigated in solid organs and limbs with very promising results [50,51,52]. Lengelé recently reported a great contribution in this field. In a recent study, he reported ECM production from human cadaveric face grafts. His protocol successfully decellularized segmental and total faces. Complex acellular facial scaffolds were obtained, simultaneously preserving a cell-friendly extracellular matrix and a perfusable vascular tree. This step will enable the further engineering of postmortem facial grafts, thereby offering new perspectives on VCA [53].

## 5. Conclusions

Encouraging results and the fact that the face transplant is the only possible way to recover certain functions and to restore the possibility of social relationships for individuals with severe facial disfigurements lead us to consider facial transplantation an important therapeutic resource and one for which there are no alternative options to date. The most recent achievements in the field of FT may be combined with cutting-edge regenerative medicine procedures and innovative immunological processing. It is paramount to build strong international networks between the world FT experts in order to achieve higher-level outcomes and reduce the complication rate. Nevertheless, the utmost caution is required in patient selection, clinical assessment, strict follow-up, and rejection management.

## Figures and Tables

**Figure 1 jcm-11-05750-f001:**
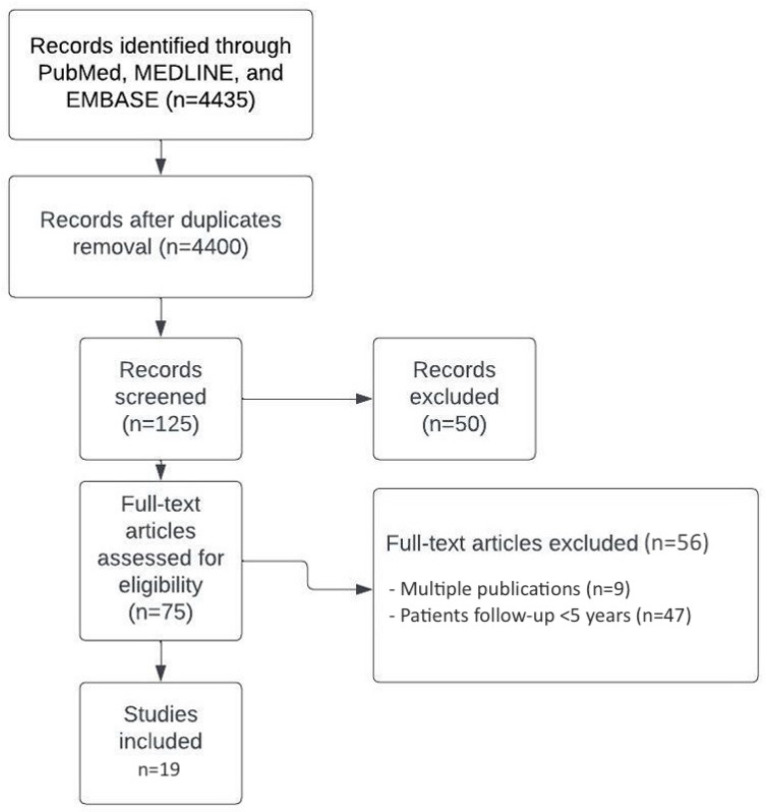
This diagram shows the flow of information through the different phases of this review.

**Figure 2 jcm-11-05750-f002:**
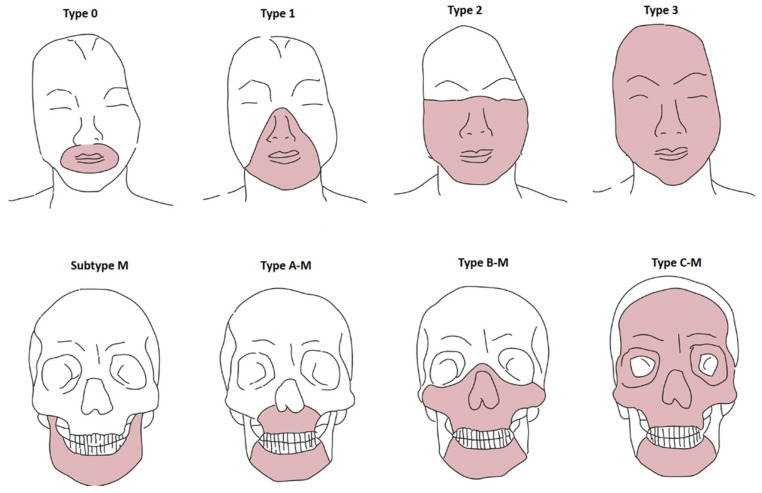
Facial tissue defect classification system schematized by the authors for facial transplantation. Soft-tissue defects (**above**) and skeletal tissue defects (**below**).

**Figure 3 jcm-11-05750-f003:**
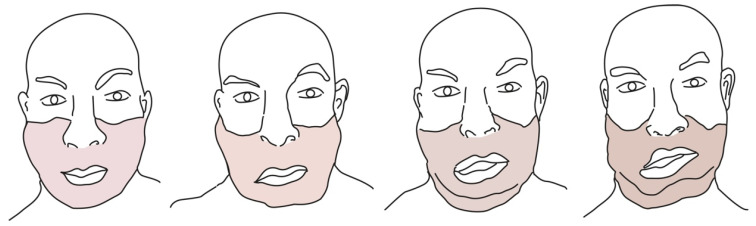
Some representative changes that can occur in FT patients, especially in those who have faced multiple acute rejection episodes. We noted, especially in patients who faced multiple acute rejection episodes, a lymphedematous aspect, sagging, asymmetry, loss of laxity, and pigmentation changes in the transplanted area.

**Table 1 jcm-11-05750-t001:** Studies included in the review.

Authors	Title	Journal	Year of Publication
Kantar RS et al. [2]	Facial Transplantation: Principles and Evolving Concepts	*Plast. Reconstr. Surg.*	2021
Diep GK et al. [3]	The 2020 Facial Transplantation Update: A 15-Year Compendium	*Plast. Reconstr. Surg. Glob. Open*	2021
Kantar RS et al. [4]	Incidence of preventable nonfatal craniofacial injuries and implications for facial transplantation	*J. Craniofac. Surg.*	2019
Chandraker A et al. [6]	The management of antibody-mediated rejection in the first presensitized recipient of a full-face allotransplant	*Am. J. Transplant.*	2014
Bharadia D et al. [7]	Role of facial vascularized composite allotransplantation in burn patients	*Clin. Plast. Surg.*	2017
Rifkin WJ et al. [8]	Achievements and challenges in facial transplantation	*Ann. Surg.*	2018
Sosin M, Rodriguez ED. [9]	The face transplantation update: 2016	*Plast. Reconstr. Surg.*	2016
Lindford AJ et al. [10]	The Helsinki approach to face transplantation	*J. Plast. Reconstr. Aesthet. Surg.*	2019
Ramly EP et al. [11]	Computerized approach to facial transplantation: Evolution and application in 3 consecutive face transplants	*Plast. Reconstr. Surg. Glob. Open*	2019
Diaz-Siso JR et al. [12]	Novel donor transfer algorithm for multi-organ and facial allograft procurement	*Am. J. Transplant.*	2017
Rifkin WJ [13]	Long-distance care of face transplant recipients in the United States	*J. Plast. Reconstr. Aesthet. Surg.*	2018
Lantieri L et al. [14]	Face transplant: Long-term follow-up and results of a prospective open study	*Lancet*	2016
Ramly EP et al. [15]	Outcomes after tooth-bearing maxilloman-dibular facial transplantation: Insights and lessons learned	*J. Oral. Maxillofac. Surg.*	2019
Khalifian S et al. [16]	Facial transplantation: The first 9 years	*Lancet*	2014
Lantieri L et al. [17]	First human facial retransplantation: 30-month follow-up	*Lancet*	2020
Kauke M et al. [18]	Full facial retransplantation in a female patient-Technical, immunologic, and clinical considerations	*Am. J. Transplant.*	2021
Fischer S et al. [19]	Functional outcomes of face transplantation	*Am. J. Transpl.*	2015
Petruzzo P et al. [20]	First human face transplantation: 5 years outcomes	*Transplantation*	2012
Morelon E et al. [43]	Face transplantation: partial graft loss of the first case ten years later	*Am. J. Transplant.*	2017

**Table 2 jcm-11-05750-t002:** Deceased Patients.

Patient	Team	Location and Date of FT	Patient (Age, Gender)	Indication	Allograft Type	Cause of Death, Time from Transplantation	AR	CR
**1**	Devauchelle, Dubernard	Amiens, France, 2005	38, woman	Animal attack	Partial	Malignancy, 8 years	Yes	Yes
**2**	Guo	Xi’an, China, 2006	30, woman	Animal attack	Partial	Non-compliance, 27 months	Yes	No
**3**	Siemionow	Cleveland, Ohio, 2008	45, woman	Ballistic trauma	Partial	Infection, 11 years and 7 months	Yes	No
**4**	Lantieri, Meningaud	Créteil, France, 2009	37, man	Third-degree burn	Partial + Hands	Sepsis (multidrug-resistant Pseudomonas), 2 months	No	No
**5**	Pomahac	Boston, Massachusetts, 2009	59, man	Electrical burn	Partial	Hepatocellular carcinoma, 10 years	Yes	Yes
**6**	Cavadas	Valencia, Spain, 2009	42, man	Osteoradionecrosis after malignancy	Partial	Malignancy (unknown)	Yes	No
**7**	Lantieri, Meningaud	Créteil, France, 2011	41, man	Ballistic Trauma	Partial	Suicide, 36 months	Yes	No
**8**	Ozkan	Ankara, Turkey, 2013	54, man	Ballistic Trauma	Full	Multiorgan deficiency resulting from progressive infectious (pulmonary and cerebellar aspergillosis) and metabolic complications, 11 months	Yes	No

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
