# Peer review of "Face Transplant: Indications, Outcomes, and Ethical Issues—Where Do We Stand?"

_jcm, 2022, doi:10.3390/jcm11195750_

Round 1

Reviewer 1 Report

I read with interest the review of La Padula et al. about face transplantation.
They  performed an extensive literature search in the face transplant field and summarized the most recent achievements and long-term clinical sequelae.
I found it interesting and well written. This is a complex and broad subject that the authors have dealt with in an exhaustive and clear way.
The English is good and the text is clear and easy to understand.

Author Response

DEAR COLLEAGUE

Thank you so much for your kind comments

I really appreciate it

Thank you for your time and patience

Kind regards

Reviewer 2 Report

The article is well structured in its form and a thorough search of articles on face transplantation has been analysed, including them fully.

The article analyses the patient from inclusion in protocols to long-term sequelae. The interest of the contents make this article a key read for those who might venture into this field.

The form and content and the English is good.

Author Response

(The authors gave the same response as above.)

Reviewer 3 Report

The authors reviewed the existing literature about face transplantation.

There are several issues with this manuscript:

1.         the review is written very superficially. the reader expects more detail in the given information, eg.:

-            line 190 "avoiding high levels of catecholamines". the authors have to be more specific and present values.

-            line 194: "Then a cast of the face is made". This is not sufficient. How is the cast made? Present details, eg. material, process, time, costs etc.

-            line 215: "surgeons will rinse the graft and inject it with a preservation solution". The same problem. Be more specific, more details.

-            line 268: "appropriate antimicrobial prophylaxis is necessary". Which and why? For how long? which flora are found more frequently?

The whole review should be revised to contain valuable information and not general statements.

2.         the quality of the review does not justify the number of authors. The authors should critically review their contributions and decide if all authors meet the ICMJE criteria. Ten authors for such a review is an unacceptably high number.

3. please explain why you applied these inclusion criteria (5-year follow-up period after FT)

Author Response

DEAR COLLEAGUE

Thank you so much for your kind comments

I really appreciate it

Thank you for your time and patience

Kind regards

  1. the review is written very superficially. the reader expects more detail in the given information, eg.:

-            line 190 "avoiding high levels of catecholamines". the authors have to be more specific and present values.

Dear colleague I have added this statement:

During organ harvesting, catecholamines are administered to maintain adequate blood perfusion (doses may vary depending on the patient's condition).

-            line 194: "Then a cast of the face is made". This is not sufficient. How is the cast made? Present details, eg. material, process, time, costs etc.

Dear Colleague thank you for this observation

We added this description (in red):

Various methods of donor face restoration are today available, most of which include different materials and molding techniques to produce donor masks to restore, to the extent possible, the preoperative appearance of the donor face. Alginate is usually used to create a mold and impression in negative of the donor’s face; this process requires approximately 30 minutes to complete. Once the alginate impression is completed, mask production continues in a separate room with colored acrylic resins that are subsequently poured into the mold. The mask is then perfected with the application of makeup by a maxillofacial prosthetic technician or anaplastologist, using a photograph of the donor. At the end of the harvesting of the facial VCA, the mask is placed on the donor under the surgeon’s supervision. The average production time for the masks is about 4 hours, and the cost of materials is $50 per mask [22].

-            line 215: "surgeons will rinse the graft and inject it with a preservation solution". The same problem. Be more specific, more details.

Dear Colleague thank you for this observation

We added this description (in red):

Static cold storage of the VCA and preservation solutions are used to reduce the risk of ischemia–reperfusion injury.

Preservation solutions are differentiated based on their Na+/K+ ratio.

High sodium is a characteristic of an extracellular-type solution (ETS) while high potassium corresponds to an intracellular-type solution (ITS). ITS maintains intracellular ion concentration

by creating a temporary cation equilibrium that compensates for the lack of active transport during

ischemia. While many different preservation solutions exist, the University of Wisconsin solution (UWS) has been used in most of the cases of facial VCA. This is an ITS solution with PH 7.4 and osmolarity 320. It contains 25 mmol/L of sodium, 125 mmol/L of potassium, 25 mmol/L of phosphate, 5 mmol/L of sulfate, 5 mmol/L of magnesium, 30 mmol/L of raffinose, 100 mmol/L of lactobionate, 5 mmol/L of adenosine, 50 mmol/L of pentafraction, 3 mmol/L of glutathione and 1 mmol/L of allopurinol.

-            line 268: "appropriate antimicrobial prophylaxis is necessary". Which and why? For how long? which flora are found more frequently?

The whole review should be revised to contain valuable information and not general statements.

Dear Colleague thank you for this observation

We added this description (in red):

Technical and anatomical aspects of facial VCA can lead to infectious complications.

Specifically, the flora colonizing donor oral and respiratory mucosa, peripheral ganglia and lymph nodes are transferred during the procedure and predispose recipients to donor-derived infections. Furthermore, immunosuppressive therapy increases the recipient’s risk of opportunistic infections.

The flora most frequently found in the nasal cavities is represented by

Corynebacteriaceae, Propionibacteriaceae and Staphylococcus, while Prevotella, Streptococcus, Veillonella, and Neisseria genera, are more commonly found in oral spaces.

Perioperative antimicrobial prophylaxis at our institution includes vancomycin, cefazolin and micafungin for at least 4 days after surgery. The perioperative antimicrobial regimen is then adjusted based on donor and recipient cultures obtained at the time of transplantation. Pneumocystis prophylaxis with trimethoprim-sulfamethoxazole and cytomegalovirus (CMV) prophylaxis with valganciclovir are given for 6 months post-facial VCA, or longer if VCA recipients

receive treatment for acute rejection.

  1. the quality of the review does not justify the number of authors. The authors should critically review their contributions and decide if all authors meet the ICMJE criteria. Ten authors for such a review is an unacceptably high number.

Dear Colleague thank you for this observation

All the authors meet the ICMJE criteria. All the authors gave a strong contribution to develop this paper.

As we have also stated in the manuscript:

Ten reviewers working in pairs sequentially evaluated the titles, abstracts and then full text of all publications identified by our searches for potentially relevant publications. A data-charting form was jointly developed by four reviewers to determine which paper to extract and use for this review.

During the submission process, all the authorship criteria have been explained to the Editorial office, and if needed I will send to the Editorial office the personal contribution of each author.

  1. please explain why you applied these inclusion criteria (5-year follow-up period after FT)

Dear colleague, 5-year follow-up was used as an inclusion criterion because we wanted to have a longer follow-up than most studies already available, and especially because of the observation that some complications (discoloration of VCA, sagging of facial structures, etc.) are more frequent 4-5 years after facial VCA.

Dear colleague, I hope you will appreciate my effort to improve the quality of this article; it was a tremendous amount of work for all our team members. I would like to thank you once again for your time and efforts in reviewing my manuscript.

Reviewer 4 Report

The paper is well written and gives an overview on the FT topic.

Please identify the contributions of the several authors. I do not see a FT program. 

Please describe the author's experience scientifically. Currently it is just anecdotal.

Please refrain from such judgmental/valued/non scientific/bloomy statements like below

- Soft-tissue and skeletal tissue defects for Facial allograft have been brilliantly classified by Eduardo D. Rodriguez et al. [2]: 

- The first Author  (SLP) had the privilege to visit this patient in 2017 few months before retransplantation,  since he was followed by the Immunology and Nephrology teams of the Henri Mondor  University Hospital - Créteil (where he received his first FT by Professor Lantieri and Professor Meningaud). When he was called to visit the patient he observed diffuse telangie tasia, dyschromia and skin sclerosis with allograft dysfunction (limited facial motion). 

The patient underwent facial retransplantation in January 2018 at the Georges-Pompidou  European Hospital (Paris), where Professor Lantieri is currently Director of the Plastic Sur gery Department. 

The first author had the honor to spent 8 years in the Plastic and Maxillofacial Surgery Department of the Henri Mondor Hospital (Créteil, France) headed by Professor JP 448 Meningaud. Some of the patients who received FT by professor Meningaud and professor  Lantieri, were followed during this period 

Professor Meningaud brilliantly answered to this question in his book [13] 

Author Response

The paper is well written and gives an overview on the FT topic.

Please identify the contributions of the several authors. I do not see a FT program. 

Please describe the author's experience scientifically. Currently it is just anecdotal.

DEAR COLLEAGUE

Thank you so much for your kind comments

I really appreciate it

Thank you for your time and patience

I understand what you mean, but Professor Meningaud performed all the 7 facial VCA at the Henri Mondor Hospital with Professor Lantieri. We are all part of his FT group, and since this is a literature review, we focused on the review, and I only added the most relevant team’s observations of the operated patients on a long term follow up.

Please refrain from such judgmental/valued/non scientific/bloomy statements like below

Dear Colleague

I totally agree with you, so these demanded corrections have been performed.

- Soft-tissue and skeletal tissue defects for Facial allograft have been brilliantly classified by Eduardo D. Rodriguez et al. [2]: 

- The first Author  (SLP) had the privilege to visit this patient in 2017 few months before retransplantation,  since he was followed by the Immunology and Nephrology teams of the Henri Mondor  University Hospital - Créteil (where he received his first FT by Professor Lantieri and Professor Meningaud). When he was called to visit the patient he observed diffuse telangie tasia, dyschromia and skin sclerosis with allograft dysfunction (limited facial motion). 

The patient underwent facial retransplantation in January 2018 at the Georges-Pompidou  European Hospital (Paris), where Professor Lantieri is currently Director of the Plastic Sur gery Department. 

The first author had the honor to spent 8 years in the Plastic and Maxillofacial Surgery Department of the Henri Mondor Hospital (Créteil, France) headed by Professor JP 448 Meningaud. Some of the patients who received FT by professor Meningaud and professor  Lantieri, were followed during this period 

Professor Meningaud brilliantly answered to this question in his book [13] 

Dear colleague, I hope you will appreciate my effort to improve the quality of this article; it was a tremendous amount of work for all our team members. I would like to thank you once again for your time and efforts in reviewing my manuscript.

Reviewer 5 Report

Dear authors,

congratulations on this interesting manuscript about the highly-specific field of face transplantation.

The manuscript is very well structured and includes all necessary information and references.

However, i strongly recommend editing of English language and style/spell check.

Some examles:

- line 61: Insert space characters before and after ">"

- line 62: "Exclusion criteria were reports with a patients follow up < 5 years". A report can not be an exclusion criterion. The follow-up < 5 years is the exclusion criterion. Therefore you should write "Exclusion criteria were a patients follow up < 5 years ......

- line 107: "partially lost", not "partially loss"

- line 184: "ORL": Please explain the abbreviation

- line 212: "the external ear do not part of the face but can be taken with"

Please correct to : "...the external ear are not a part of the face but can be included in the harvest".

- line 217: " Currently, it is rare for the total ischemia time exceeds four hours"

Please correct to: "Currently, it is rare that the total ischemia time exceeds four hours"

- line 233: "veil"? Do you mean vein?

Author Response

Dear authors,

congratulations on this interesting manuscript about the highly-specific field of face transplantation.

The manuscript is very well structured and includes all necessary information and references.

DEAR COLLEAGUE

Thank you so much for your kind comments

I really appreciate it

Thank you for your time and patience

However, i strongly recommend editing of English language and style/spell check.

Dear Colleague

The paper has been fully reviewed by our colleague Dr Pizza a native English speaker

Some examles:

- line 61: Insert space characters before and after ">"

Dear Colleague This error has been corrected

- line 62: "Exclusion criteria were reports with a patients follow up < 5 years". A report can not be an exclusion criterion. The follow-up < 5 years is the exclusion criterion. Therefore you should write "Exclusion criteria were a patients follow up < 5 years ......

Dear Colleague This error has been corrected

- line 107: "partially lost", not "partially loss"

Dear Colleague This error has been corrected

- line 184: "ORL": Please explain the abbreviation

Dear Colleague This error has been corrected

- line 212: "the external ear do not part of the face but can be taken with"

Please correct to : "...the external ear are not a part of the face but can be included in the harvest".

 Dear Colleague This error has been corrected

- line 217: " Currently, it is rare for the total ischemia time exceeds four hours"

Please correct to: "Currently, it is rare that the total ischemia time exceeds four hours"

 Dear Colleague This error has been corrected

- line 233: "veil"? Do you mean vein?

Dear Colleague  I mean palatine veil, This error has been corrected

Dear colleague, I hope you will appreciate my effort to improve the quality of this article; it was a tremendous amount of work for all our team members. I would like to thank you once again for your time and efforts in reviewing my manuscript.

Round 2

Reviewer 3 Report

The manuscript has not greatly improved.